# Novel Insights and Genomic Characterization of Coral-Associated Microorganisms from Maldives Displaying Antimicrobial, Antioxidant, and UV-Protectant Activities

**DOI:** 10.3390/biology14040401

**Published:** 2025-04-11

**Authors:** Fortunato Palma Esposito, Andrea López-Mobilia, Michael Tangherlini, Vincenza Casella, Alessandro Coppola, Giulia Varola, Laura Vitale, Gerardo Della Sala, Pietro Tedesco, Simone Montano, Davide Seveso, Paolo Galli, Daniela Coppola, Donatella de Pascale, Christian Galasso

**Affiliations:** 1Department of Ecosustainable Marine Biotechnology, Stazione Zoologica Anton Dohrn, Via Ammiraglio Ferdinando Acton 55, 80133 Naples, Italy; andrea.lopez-mobilia@szn.it (A.L.-M.); vincenza.casella@szn.it (V.C.); alessandro.coppola@szn.it (A.C.); giulia.varola@szn.it (G.V.); laura.vitale@szn.it (L.V.); gerardo.dellasala@szn.it (G.D.S.); daniela.coppola@szn.it (D.C.); 2Department of Research Infrastructures for Marine Biological Resources, Stazione Zoologica Anton Dohrn, Fano Marine Centre, Viale Adriatico 1, 61032 Fano, Italy; michael.tangherlini@szn.it; 3Department of Chemical, Biological, Pharmaceutical and Environmental Sciences, University of Messina, Viale F. Stagno d’Alcontres, 31, 98166 Messina, Italy; 4Department of Earth and Environmental Science, University of Milano Bicocca, Piazza della Scienza 1, 20126 Milano, Italy; simone.montano@unimib.it (S.M.); davide.seveso@unimib.it (D.S.); paolo.galli@unimib.it (P.G.); 5Marine Research and High Education Center (MaRHE Center), Magoodhoo Island, Faafu Atoll 12030, Maldives; 6Department of Ecosustainable Marine Biotechnology, Calabria Marine Centre, CRIMAC (Centro Ricerche ed Infrastrutture Marine Avanzate in Calabria), Stazione Zoologica Anton Dohrn, C. da Torre Spaccata, 87071 Amendolara, Italy; christian.galasso@szn.it

**Keywords:** climate change, coral reefs, Maldives, coral-associated microorganisms, bioactivity, genome annotation, environmental biotechnology, marine natural products, coral restoration

## Abstract

Climate change produces strong impacts on marine ecosystems, especially in tropical environments, where deleterious effects are largely evident. Rising in temperatures and ocean acidification are increasing the rate of coral bleaching and death. Coral-associated bacteria play a fundamental role in the defense and protection of coral from climate change-related stressors. An innovative strategy of coral restoration to counteract the detrimental effect of climate change on reefs is a boost in coral microbiome through enrichment with bioactive microbial strains. This research paper characterizes coral microbiomes of two Maldivian species, *Porites lobata* and *Acropora gemmifera*, trying to highlight similarities and differences between corals according to their depths and growth stages. All coral samples were then used to isolate bacteria and study their bioactivity, identifying the four most active strains (including a novel species) with antimicrobial, antioxidant, and UV-protective effects. The whole genomes of bioactive strains were examined, searching for functional genes relevant for the beneficial effect in the bacteria–coral relationship and clusters encoding for secondary metabolites of biotechnological interest. These findings reinforce the importance of investigating coral-associated bacteria to find new solutions for future environmental and human health applications.

## 1. Introduction

Coral reefs are considered among the largest hot spot of marine chemical and genetic diversity [1] and one of the most productive ecosystems, offering ecological, economic, and social benefits. The health status of some coral barriers is correlated with human impacts due to local populations whose economy is based on coastal activities. The Republic of Maldives is an appropriate example, being composed of more than 1100 small islands in the Indian Ocean. These islands form atolls and are characterized by a large diversity of marine habitats, such as shallow and deep coral reefs, lagoons, and mangroves [2]. Unfortunately, these ecosystems face escalating threats from human activities, including climate change, land reclamation, overfishing, and pollution [3,4,5], with direct effects on coral vitality and reef stability [6,7,8]. To counteract the alarming degradation of coral reefs, restoration efforts have become a critical strategy [9,10]. Techniques such as coral transplantation show considerable promise [11]. However, their long-term effectiveness strongly relies on a deeper understanding of the intricate relationships between corals and their associated microorganisms [12].

At the microscopic scale, coral reefs are composed of millions of polyps that live in symbiosis mainly with dinoflagellates of the *Symbiodiniaceae* (photosynthetic microalgae) family [13]. In addition to the coral–microalgae relationship, the coral holobiont is composed of archaea, bacteria, fungi, and viruses, forming a small ecosystem [14,15,16]. This associated microbial community rapidly responds to environmental stimuli and signals (e.g., heat waves, acidification, light intensity, pollution), changing its composition and metabolic activities [17]. Recent efforts have focused on taxonomic characterization of coral-associated microorganisms (CAMs), creating and implementing databases [18] to support a better understanding of the role of bacteria in symbiosis [19,20]. Certain bacterial symbionts may offer benefits to corals, playing a crucial ecological role, for instance, in food supply, participating in the carbon, nitrogen, and sulfur cycles; in thermal stress mitigation; and in defense strategies against pathogens and predators [21,22,23]. In this context, the term “beneficial microorganisms for corals” (BMCs) has been proposed to identify symbionts (e.g., bacteria) that promote coral health and could be exploited for coral restoration purposes [24]. The application of BMCs, such as probiotic inoculation, has demonstrated potential as a therapeutic approach for corals experiencing environmental stress events that typically induce dysbiosis (characterized by a detrimental imbalance in the microbiome) [25]. Following this principle, numerous microbe-based therapies have been exploited for coral restoration purposes, such as the use of prebiotics, postbiotics, and bacterial adaptation. Other microbial-based approaches include microbiome transplantation and the regulation of bacterial populations through phage therapy [26]. These microbial therapies leverage the mutualistic interactions between corals and their associated microorganisms, enabling the delivery of selected microbes, microbial products, or microbial substrates.

For example, beneficial effects can be mediated through bacterial secretion/production of bioactive compounds (e.g., antimicrobial, antioxidant, and photoprotective molecules), making corals more resilient to environmental changes and stressors [20,24,27,28].

Indeed, a plethora of marine natural products have been discovered from CAMs, displaying bioactivities useful for environmental biotechnology and possible pharmaceutical developments [29]. Concerning environmental applications, CAMs could protect corals from bleaching, a phenomenon mainly driven by elevated temperatures and high solar irradiance, and closely linked to the overproduction of reactive oxygen species (ROS) and reactive nitrogen species (RNS) [30,31]. In this regard, CAMs possess scavenging enzymes, including catalase and superoxide dismutase, and non-enzymatic antioxidants (e.g., mannitol, glutathione, and carotenoids) that could mitigate ROS damage [32]. The development of stress-tolerant microbial strains as probiotics is being actively explored to strengthen coral health under challenging conditions [24]. These interventions align with the emerging concept of “assisted microbiome engineering”, which involves tailoring microbial communities to deliver specific benefits to the coral host and mitigate the effects of climate change [25,33].

Regarding pharmaceutical applications, the chemical ecology of coral-associated bacteria can drive the discovery of interesting lead compounds for further drug development. For instance, bacteria defend corals from pathogen infections, producing antimicrobial and antifouling compounds [34]. Bacteria can also produce sunlight screens able to form a barrier for the deleterious effect of UV rays [35].

However, the relevance of CAMs should be further investigated to better expand the concept of BMCs through (i) the elucidation of coral microbiome compositions and distribution, (ii) isolating associated microbes in pure cultures, and (iii) exploring the biotechnological potential through the discovery of novel natural products produced by the isolated strains. As a matter of fact, most of the knowledge about CAMs has been generated through metabarcoding or metagenomic analysis [20], while few studies involve the isolation of microorganisms and, even less, the evaluation of their biotechnological activities [36]. The combination of both culture-dependent and metabarcoding approaches could provide more informative data, increasing the current knowledge about CAMs and their importance in coral conservation.

This research started from the scientific hypothesis that the bio-chemical complexity of the symbiotic relationship between corals and microorganisms can be a rich source of bacteria already chemically adapted to face climate change-related stressors. Thus, this study aimed to use a holistic approach to investigate (i) how and whether different depths and growth stages drive the structure of coral microbiomes, (ii) the bioactivity of prokaryotes isolated from corals, and (iii) the hidden genomic potential of selected microbes to better elucidate their relationship with corals. Thus, metabarcoding analysis was conducted to characterize the microbial community associated with *Porites lobata* and *Acropora gemmifera*, collected from shallow and deep reefs and at different growth stages. The selection of these two coral species is primarily due to their abundance being representative of the Maldivian reef ecosystem and their relevant ecological role in that area. Moreover, they exhibit notable morphological differences in terms of tissue and skeleton thickness, as well as at the molecular, physiological, and metabolic levels, displaying a different susceptibility and tolerance to environmental stress, particularly the stressors associated with climate change that lead to coral bleaching [37], making their microbiome worth being investigated. Coral-associated bacteria were isolated, and their biological activities were assessed in terms of antimicrobial, antioxidant, and photoprotective effects. The genomes of the four most promising bacteria were analyzed, searching for relevant genes that could be involved in the beneficial coral–microbe relationship. Thus, the final aim of this work is the identification of promising bacteria with associated information on bioactivity and genome potential that could lay the basis for future coral restoration strategies, together with the identification of potential new marine sources for human health applications.

## 2. Materials and Methods

### 2.1. Sampling Site and Coral Collection

Sampling activities were conducted in the coral reef of Faafu Atoll (Figure 1A), Republic of Maldives, in February 2024, using the Marine Research and Higher Education (MaRHE) Center as a logistics base (https://marhe.unimib.it/, accessed on 27 February 2025), which is located on Magoodhoo Island (Faafu, Maldives). In particular, the dive site called “Wallstreet” (3°6′52.08″ N; 72°59′4.48″ E, see Figure 1B) was used, which is located in the coral reef in front of the island of Bileiydhoo and is approximately 5 km from MaRHE Center. Wallstreet is characterized by high coral coverage and diversity, interspersed with small sand pockets and distinguished by a broad, coral-rich flat that extends 20 m wide. A steep wall descends to depths of around 30 m, gradually sloping to 35 m before transitioning to the sandy seabed. This vertical descent features various crevices, caves, and overhangs that provide essential shelter for a diverse array of marine life. The site is typically influenced by medium to slow currents, fostering optimal conditions for the proliferation of diverse coral and fish communities.

Coral colonies were identified and simultaneously sampled at approximately 10 a.m. to ensure that all replicates were subjected to the same environmental conditions (e.g., light, temperature, current), following the scheme illustrated in Figure 1C. In detail, two sampling depths were identified at 5 m (shallow) and 17 m (deep). At both depths, small (≤10 cm in diameter, juvenile colonies) and large colonies (≥30 cm in diameter, adult colonies) of *Porites lobata* (massive growth form) and *Acropora gemmifera* (branched growth form) were sampled in biological triplicate. Therefore, 6 colonies (3 small and 3 large) per depth (2, shallow and deep) per species (2) were collected, for a total of 24 colonies. The diameter of each colony was measured using calipers and a measuring tape considering the longest axis, as previously reported [35]. Fragments of coral colonies (approximately 20 cm^2^) were collected by hammer and chisel and plier by SCUBA diving and stored in sterile bags in seawater.

Water and sediments were collected close to the coral colonies. Sediments were collected in triplicate using 50 mL tubes from sand pockets, and seawater (3 L per replicate) was collected using sterile bags. All the samples were quickly stored in a refrigerated box until they reached the laboratory. Coral fragments were washed with filtered seawater, reduced into smaller pieces with sterile hammer and scalpel, placed in 50 mL tubes containing RNAlater (for DNA extraction) (Invitrogen™ AM7021, Waltham, MA, USA) and filtered seawater (for bacterial isolation), and stored at 4 °C. Seawater was removed from tubes with sediments and then filled with RNAlater and stored at 4 °C. Seawater samples were vacuum-filtered using sterile nitrocellulose filters (0.2 μm porosity, 47 mm diameter; WHA10401770, Sigma Aldrich, St. Louis, MO, USA), stored in 2 mL tubes filled with RNAlater, and stored at 4 °C.

### 2.2. DNA Extraction and Sequencing

Sediments and corals (300 mg) were extracted using DNeasy PowerSoil Pro Kit (Qiagen, cat. n. 47014, Hilden, Germany). Coral samples (comprising a mixture of tissue, mucus, and skeleton) were fragmented using a sterile pestle and mortar prior to DNA extraction. Seawater filters were extracted using Dneasy PowerWater Kit (Qiagen, cat. n. 14900-100-NF, Hilden, Germany). Seawater filters were reduced to smaller pieces using sterile scalpels before DNA extraction. DNA concentrations were assessed using the NanoDrop microvolume spectrophotometer (ThermoFisher Scientific) and the Qubit™ dsDNA Quantification Assay Kit (High Sensitivity, cat. N. Q32851, Waltham, MA, USA). Size range and DNA integrity were evaluated on agarose gel electrophoresis (0.8%). Next-generation sequencing was performed by the external service Eurofins Genomics (https://eurofinsgenomics.eu/, accessed on 27 February 2025), Ebersberg, Germany), and the amplification was carried out using the primer TACGGGAGGCAGCAG (forward) and CCAGGGTATCTAATCC (reverse) [38,39].

### 2.3. Microbial Community Characterization

Raw sequences were analyzed through the QIIME2 pipeline (version 2022.2; https://qiime2.org/, accessed on 22 January 2025) [36]. The cutadapt plugin was used to remove primer sequences from the read pairs with standard parameters [40]. Trimmed paired-end sequences were analyzed through the DADA2 procedure, thus removing chimeric sequences, correcting sequencing errors, and generating Amplicon Sequence Variants (ASVs) using standard parameters [41]. Taxonomy was assigned using the reference database SILVA (version 138 nr SSU) [42]. Taxonomic affiliation was performed through the VSEARCH plugin after trimming the reference dataset to the region amplified by the chosen primers [43]. For a proper comparison among samples for alpha- and beta-diversity analyses, as well as for taxonomic investigation of the assemblage composition, the ASV table was rarefied to a common sampling depth of 23,000 sequences [44]. After rarefaction, data analysis was performed with R (version v4.2; https://www.r-project.org/, accessed on 22 January 2025) within the Rstudio program using the *Microeco* package [45].

### 2.4. Bacterial Isolation and Preliminary Molecular Identification

Coral samples were resuspended in filtered seawater (FSW) and agitated on vortex in a 50 mL tube for 30 min, allowing the release of coral-associated bacteria in liquid solution. Then, 1 mL of enriched FSW was collected from the 24 coral samples and serially diluted to 10^−7^. Each dilution was spread on two different solid growth media and incubated at 28 °C up to 3 months, allowing for the development of slow-growing strains. Growth media used for the experiments were prepared as follows:

Marine broth modified (MBmod—19.4 g NaCl, 8.8 g MgCl, 20 g bacteriological peptone, 3.24 g NaSO_4_, 1.8 g CaCl_2_, 1 g yeast extract, 0.55 g KCl, 0.16 g NaHCO_3_, 0.10 g Fe(III) citrate, 0.08 g KBr, 0.034 g SrCl_2_, 0.022 g H_3_BO_3_, 0.008 g Na_2_HPO_4_, 0.004 g Na-silicate, 0.0024 g NaF, 0.0016 g NH_4_NO_3_, pH 7.6 at 25 °C, in 1 L of distilled water); M1 medium (10 g starch, 4 g yeast extract, 2 g peptone, in 1 L of FSW).

Solid media were prepared by adding 17.0 g of bacteriological agar to 1 L of liquid medium. CaCO_3_ (2 g) was used to supplement all media. Moreover, 50 mg/L of nystatin and 25 µg/mL nalidixic acid were added to M1 medium to inhibit the growth of fungi and reduce the presence of fast-growing strains, respectively. After the incubation period, single colonies were selected based on their different morphology and colors. Then, they were sub-cultured to check for contaminations, inoculated in liquid, and stored at –80 °C in the presence of 20% glycerol.

The phylogenetic affiliation of bacterial isolates was performed through 16S rRNA gene amplification and analysis. The “freeze and thaw” method was used to obtain bacterial lysate. Briefly, colonies from each isolate were picked, transferred in a 0.5 mL tube, dissolved in 100 μL of sterile water, mixed, and frozen at −20 °C for 1 h. Thereafter, the frozen tubes were incubated for 15 min at 99 °C in a thermoshaker and centrifuged for 10 min at 8900 rcf; then, PCR was carried out in a total volume of 40 μL, containing a DNA sample, water, 25 μL of PCR Master Mix 2× ExtraWhiteTaq (a ready-to-use solution containing TaqPol, buffer, MgCl2, and dNTPs), and 0.2 μM of both primers, 27F (forward, seq: 5′-AGAGTTTGATCCTGGCTCAG-3′) and 1492R (reverse, seq 5′-GGTTACCTTGTTACGACTT-3′). The reaction conditions used were an initial denaturation step (93 °C for 2 min), 30 amplification cycles (92 °C for 30 s, 55 °C for 30 s, and 72 °C for 1 min), and a final extension step of 5 min at 72 °C. PCR products were then purified (by NucleoSpin Gel and PCR Clean-up Kit, Macherey-Nagel, Ping-Tung, Taiwan), sequenced by a commercial sequencing service (Mix2Seq Service, Eurofins Genomics, Ebersberg, Germany), and submitted to BLASTn (NCBI) and the EzBioCloud tool for the preliminary phylogenetic affiliation [46].

### 2.5. Bacterial Cultivation and Extraction

Ten bacterial strains selected based on their different taxonomy (at genus level) were inoculated in 3 mL of liquid MBmod in sterile bacteriological tubes. After 48 h of incubation at 28 °C at 180 rpm, the preinoculum was used to inoculate 250 mL of MBmod in a 1 L flask, at a starting concentration of 0.05 OD_600_/mL (optical density measurements at 600 nm). The flasks were incubated at 28 °C with 150 rpm constant shaking for 4 days and 9 days (for the slow-growing strains). Then, the cultures were centrifuged at 7500 rcf at 4 °C for 20 min, and the obtained exhausted broth was extracted with two volumes of EtOAc, while the intracellular material was extracted by incubating the pellet with 3 volumes of MeOH for 3 h in agitation. Successively, both organic phases were collected and evaporated with a rotary evaporator (Buchi R-100, Büchi Labortechnik AG, Postfach, Switzerland) to obtain total extracts (extracellular and intracellular extracts). Finally, all the extracts were weighted, and part of them dissolved in DMSO or MeOH at 50 mg/mL for biological assays and stored at −20 °C.

### 2.6. Functional Screening of Extracts from the Isolated Strains

Antimicrobial, antioxidant, and UV protectant assays were performed to assess the bioactivity of the two groups of extracts (intracellular and extracellular metabolites) obtained from the isolated strains.

#### 2.6.1. Antimicrobial Assay

The antimicrobial activities were assessed on a panel of both human and fish/coral pathogens, including *Staphylococcus aureus* ATCC 6538p™, methicillin-resistant *S. aureus* (MRSA), quinolone-resistant *S. aureus* (QRSA), vancomycin-resistant *S. aureus* (VRSA), *Vibrio anguillarum* ATCC 19264™, *Photobacterium damselae* subsp. *Piscicida* ATCC 51736™, *Escherichia coli* ATCC 10536™, macrolide-resistant *S. aureus* (MRSA), *Listeria monocytogenes* MB 677, *Pseudomonas aeruginosa* PAO1, and *Candida albicans* ATCC 76485™. Each strain was plated on MB, LB, TSB, or YPD agar plates and incubated overnight at 28 °C or 37 °C, depending on cultivation requirements. Then, 2–3 fresh colonies were transferred in 3 mL of medium (MB, LB, TSB, or YPD) and incubated in agitation overnight at 28 °C or 37 °C. Finally, a bacterial concentration of about 5 × 10^5^ CFU/mL was used for the liquid inhibition assay [47]. To evaluate the antimicrobial potential of the extracts, samples were placed in 96-well microtiter plates at an initial concentration of 0.5 mg/mL and serially diluted (two-fold dilutions) using the respective medium. Wells containing no compound represented the negative control. DMSO was used as positive control to determine the effect of solvent on cell growth. Briefly, 8 µL of each sample was dispensed into wells with 200 µL of appropriate medium and twofold serially diluted. Then, 100 µL of bacterial suspensions were inoculated into the broth (~5 × 10^4^ CFU/well) and incubated statically for 18–24 h at 28 °C or 37 °C, and the growth was measured by a Cytation3 Plate Reader (Biotek, Winoosky, VT, USA) monitoring the absorbance at 600 nm.

#### 2.6.2. DPPH Assay

The capacity of intracellular and extracellular extracts to scavenge the stable radical DPPH (diphenyl-1-picrydrazyl) was assessed [48] in 96-well microplates. The extracts were dissolved in MeOH at an initial concentration of 50 mg/mL, and 4 µL of each extract was added to wells containing 196 µL of MeOH. Then, a 2-fold serial dilution was performed for all the extracts. The concentrations tested were 500, 250, 125, 62.5, 31.25, and 15.62 µg/mL. Finally, 100 µL of a 0.2 mM methanolic solution of DPPH was added to all the wells. Ascorbic acid (AA) was used as positive control and MeOH was used as negative control (0% of activity). The absorbance was read at 517 nm using a plate reader (Synergy HT Multi-Mode Microplate Reader, BioTek Instruments, EUA, Winooski, VT, USA).

The scavenging activity was expressed using the following equation:Scavenging activity (%) = [(Ab control − Ab sample)/Ab control] × 100
where Ab control is the absorbance of the DPPH solution, and Ab sample is the absorbance of the sample mixed with DPPH. Data are the mean values of three independent experiments performed in three technical replicates. ANOVA ordinary 2-way Dunnett’s multiple comparisons was used to highlight statistical differences between samples and negative control: 0.1234 (ns), 0.0332 (*), 0.0021 (**), 0.0002 (***), <0.0001 (****).

#### 2.6.3. ABTS Assay

The scavenging activity of the ABTS radical was determined according the method provided by Ahn [49], with slight modifications, and performed in 96-well microplates. Briefly, an ABTS stock solution (7 mM ABTS, in water) was mixed with 2.45 mM K_2_S_2_O_8_ (final concentration) and incubated in a dark room for 18 h at room temperature, allowing ABTS radical cation generation. Then, the ABTS was mixed with ddH_2_O to create the working solution, with an absorbance of 0.7 ± 0.05 at 734 nm. The concentrations tested were 500, 250, 125, 62.5, 31.25, and 15.62 µg/mL. Each sample (4 μL at 50 mg/mL) was added to 196 μL of ABTS working solution, and a 2-fold serial dilution was performed for all the extracts; then, samples were stored in the dark at 27 °C for 7 min. The absorbance was read at 734 nm using a plate reader (Synergy HT Multi-Mode Microplate Reader, BioTek Instruments, EUA, Winooski, VT, USA).

The scavenging activity was expressed using the following equation: Scavenging activity (%) = [(Ab control − Ab sample)/Ab control] × 100
where Ab control is the absorbance of the ABTS solution, and Ab sample is the absorbance of the sample mixed with ABTS. Trolox was used as positive control and only ABTS solution as negative control (0% of activity). Data are the mean values of three independent experiments performed in three technical replicates. ANOVA ordinary 2-way Dunnett’s multiple comparisons was used to highlight statistical differences between samples and negative control: 0.1234 (ns), 0.0332 (*), 0.0021 (**), 0.0002 (***), <0.0001 (****).

#### 2.6.4. *In Vitro* Cytotoxicity and UV-Screen Assay

The immortalized human epidermal keratinocyte cell line (HaCaT, cat. n° SCCE020, Sigma-Aldrich, Darmstadt, Germany) was cultured at 37 °C in a humidified atmosphere of 95% air and 5% CO_2_ in DMEM high-glucose medium (ATCC, 30-2002™) supplemented with 10% fetal bovine serum (FBS, ATCC, 30-2020™, Manassas, VA, USA) and 1% penicillin/streptomycin (ATCC, 30-2300™). Cells were seeded at a density of 2 × 10^4^ cells/well in 96-well plates and incubated for 24 h for the assessment of cytotoxicity and the UV-protective effect.

Cells were treated with 100 and 200 µg/mL of each extract for 24 h to assess their cytotoxicity. Extracts that did not show a cytotoxic effect were used for the UV-screen assay, as reported below.

For each 96-well plate, a maximum of 60 inner wells were seeded to ensure each well was fully irradiated [50]. Before UVC irradiation, cells were pre-treated with intracellular and extracellular bacterial extracts (100 and 200 µg/mL) in PBS (Corning, cat. n° 21-040-CM) for 1 h at 37 °C. (Uvinul^®^ MC 80, BASF, Ludwigshafen, Germany), was used as a positive control due to its UV protective property. After the pre-treatment, the HaCaT cells were exposed to UVC radiation for 30 s by placing the 96-well plates, without the lid, inside the laminar flow hood, directly underneath and at a distance of 55 cm from the UVC lamp (germicidal lamp TUV T8 GL15, PHILIPS, with 5 W UVC radiation and peak at 254 nm), as previously reported [50]. After UVC exposure, the bacterial extracts in PBS were removed and replaced with fresh DMEM supplemented with the same extracts. After 24 h, the cell viability was assessed by the MTT assay to define the cytotoxicity and protective effects of the extracts on the UVC-irradiated HaCaT cells. More in detail, 10 µL of MTT (5 mg/mL) was added to each well. After 3 h of incubation at 37 °C, the well’s content was removed, and the isopropanol was added to dissolve formazan crystals. Absorbance was measured by a microplate reader (Synergy HT Multi-Mode Microplate Reader, BioTek Instruments, EUA, Winooski, VT, USA) at 570 nm. For cytotoxicity, cell viability was expressed as a percentage of viable cells, calculated as the ratio between the mean absorbance of each sample and the mean absorbance of the control. The UV-protective effect was expressed as a percentage of cell viability in the presence of the tested samples, calculated as the ratio between the UV-exposed and non-exposed cells. A Student’s *t*-test was used for statistical analysis, comparing the means of the two groups (untreated cells vs. treated cells for cytotoxicity assay; Uvinul^®^ MC 80 treated cells vs. total extracts treated cells for UV-screen assay) to determine significant differences.

#### 2.6.5. Whole Genome Sequencing and Annotation

Genomic DNA (1 μg) from the strains *Pseudoalteromonas* sp. 39, *Streptomyces* sp. 79, *Microbacterium* sp. 92, and *Micromonospora* sp. 93 was sequenced using the Illumina NovaSeq 6000 platform by Eurofins Genomics (Ebersberg, Germany) to obtain the whole genome sequence. Sequenced paired-end reads were processed with the Trimmomatic tool [51] for adapter removal and quality trimming, followed by validation with FastQC (http://www.bioinformatics.babraham.ac.uk/projects/fastqc/; accessed on 5 December 2024). The *de novo* genome assemblies were obtained with SPAdes (v.15.4) [52]. To remove potentially spurious contigs and scaffolds, a round of binning though MetaBat2 [53] was carried out on each genome without providing abundance estimations prior to the binning. The CheckM2 tool [54] was then employed to obtain completeness and redundancy estimates on the four assembled genomes. FastANI [55] was utilized to identify potentially similar genomes within the Ocean Microbiomics database [56] and within the OceanDNA MAG catalog [57]; genomes found within these databases were added to the later phylogenomic affiliation step. Cleaned genome data were then annotated with RAST [58] and PROKKA [59] to obtain a broad coverage of the genetic repertoire of each strain and further annotated with DRAM [60] through the KBase webserver [61] to gather information on the metabolic potential of each strain and on the presence/absence of carbohydrate-active enzymes (CAZy). Taxonomic inferences were carried out using the GTDBtk tool with the GTDB v207 [62] on the in-house HPC server at Stazione Zoologica di Napoli “Anton Dohrn”. To further characterize the potential resistances of each strain, we compared the gene sequences resulting from the PROKKA annotation with the MEGARes database v3 [63] using usearch v10 [64] and only keeping matches with a bitscore higher than 1000. Subsequently, the genome mining tool antiSMASH 7.0 web server (http://antismash.secondarymetabolites.org, accessed on 5 December 2024 was applied to identify and annotate the biosynthetic gene clusters (BGCs) of the whole genome. Phylogenomic analyses were carried out with the GToTree tool using Gammaproteobacteria-related HMMs for genome 39 and Actinobacteria-related HMMs for the other strains [65] and adding related genomes found within the OM and OMC databases.

## 3. Results

### 3.1. Comparative Analysis and Isolation of Coral-Associated Microorganisms

Corals at two different growth stages (≤10 cm and ≥30 cm) and two depths (5 and 17 m) were collected to characterize the associated microbiome. Samples were identified as “small” and “large”, and “shallow” and “deep” colonies, respectively.

A total of 4635 ASVs were found across the entire dataset after rarefaction to 23,000 sequences. All samples from *A. gemmifera* colonies consistently showed the lowest amplicon sequence variant (ASVs) richness, ranging from 44 to 119 ASVs (Figure 2A). Higher richness values were observed in *P. lobata* samples, which ranged from 176 to 587 ASVs. The highest values were found for seawater samples (up to 1449 ASVs), followed by sediments (up to 931 ASVs). Beta-diversity analysis showed that coral samples formed two separate clusters according to their source species (Figure 2B), while seawater and sediment samples clustered together (both deep and shallow samples).

Taxonomic analyses showed that prokaryotic assemblages associated with the two corals were different from those of retrieved from seawater or sediment samples (Figure 3). In shallow reefs, large colonies of *P. lobata* were characterized by the presence of *Desulfovibrionaceae* (28.3%) and *Cyclobacteriaceae* (28.7%), whereas small colonies were mainly constituted by *Endozoicomonadaceae* (34.4%) and *Cellvibrionaceae* (20.8%). Different results were observed in deep water samples for *P. lobata*, where large colonies were characterized by the presence of *Rhizobiaceae* (38.2%) and *Microbacteriaceae* (19.7%), whereas small colonies showed a predominance of *Microtrichaceae* (22.3%) and *Nitrosococcaceae* (6.6%).

Microbiomes associated with *A. gemmifera* collected from shallow reefs showed an abundance of *Endozoicomonadaceae* (61.7% for large colonies and 87.5% for small colonies). The abundance of this family strongly decreased in deep water, where *Endozoicomonadaceae* constituted only 5.2 and 0.7% of large and small colony microbiomes, respectively, with higher frequencies of rare prokaryotic taxa. In the corals in the surrounding environment, the predominant families in shallow waters were NB1-j (9.1%) and *Woeseiaceae* (7.3%), while in deep waters they were *Rhodobacteraceae* (13.0%) and *Woeseiaceae* (5.9%). Sediments showed a similar assemblage of seawater microbiomes, as they were composed of 9.8% of *Rhodobacteraceae* and 6.9% *Woeseiaceae* in the shallow reef, while deep samples were characterized by the presence of NB1-j (13.1%) and *Kiloniellaceae* (9.7%). A complete dataset of relative abundances for each experimental group is reported in Appendix A.

About 50 strains were isolated from the different coral samples, with 37 strains from *A. gemmifera* and 18 from *P. lobata*. Among the isolates, the most morphologically different strains were selected and identified based on 16S rRNA genes. The identified strains belonged to the *Exiguobacterium* (strain 12), *Priestia* (strain 24), *Pseudoalteromonas* (strain 39), *Vibrio* (strain 40), *Saccharopolyspora* (strain 55), *Bacillus* (strain 71), *Metabacillus* (strain 72), *Streptomyces* (strain 79), *Microbacterium* (strain 92), and *Micromonospora* (strain 93) genera. Among the isolates, some industrially relevant Actinomycetota were retrieved, which emerged after a long incubation period (>45 days), such as *Streptomyces* and *Micromonospora*, due to, especially for the latter, their slow growth rate. The bioactivity and the genomic potential of the isolated bacteria were investigated, exploring relevant properties both for coral restoration purposes and for biotechnological applications.

### 3.2. Bioactivity Evaluation

The 10 selected strains were cultivated in MBmod (250 mL). Biomasses were separated from exhausted broth, and both were extracted by organic solvents (using MeOH and EtOAc, respectively). All the extracts were evaluated for different bioactivities, including (i) antimicrobial activity to identify strains able to inhibit microbial pathogens, (ii) relevant antioxidant activity in response to oxidative stress, and (iii) UV protection to identify bacteria able to cope with high solar irradiation.

#### 3.2.1. Antimicrobial Activity

A panel of 11 pathogens was implemented as target strains, including fish and multidrug-resistant human pathogens. Among the 10 selected CAMs, the extracellular extracts generated from *Pseudoalteromonas* sp. 39, *Streptomyces* sp. 79, *Microbacterium* sp. 92, and *Micromonospora* sp. 93 displayed the highest bioactivity (Table 1). *Streptomyces* sp. 79 was able to inhibit 8 out of the 11 target strains, including *Candida albicans*, showing the strongest effect towards *Photobacterium damselae* subsp. *piscicida* (a fish pathogen), with an MIC value of 0.0098 µg/mL. Similarly, *Microbacterium* sp. 92 and *Micromonospora* sp. 93 inhibited 7 out of 11 pathogens, with MIC values ranging from 125 to 0.3 µg/mL. *Pseudoalteromonas* sp. 39 instead showed potent antimicrobial activity (MIC 0.031 µg/mL) against *Vibrio anguillarum* (a fish and coral pathogen) and *P. damselae* (MIC 30 µg/mL). Notably, no extracts were reported to have an effect towards *Staphylococcus aureus* vancomycin resistant or *Pseudomonas aeruginosa*, while only a weak inhibition was displayed by *Streptomyces* sp. 79 against *Escherichia coli*. Searching for secreted molecules, intracellular extracts have not been tested for the antimicrobial activity.

#### 3.2.2. Antioxidant Activity

Two antioxidant assays (DPPH and ABTS) were applied, allowing for the detection of natural compounds with different scavenging mechanisms [66]. All extracellular extracts showed dose-dependent activity (Figure 4A,B). The extracts derived from strains 39, 92, and 93 showed the most promising results. In particular, the extract derived from *Pseudoalteromonas* sp. 39 displayed the highest antioxidant effect. In fact, this extract in the DPPH assay (Figure 4A) exerted a comparable effect to ascorbic acid for four of the six concentrations tested (500, 250, 125, 62.5 µg/mL). In the ABTS assay (Figure 4B), the same strain showed a dose-response effect, with a proportional decrease in antioxidant activity with concentration reduction. In both assays, the bioactivity of strain 39 decreased to below 10% only at the lowest concentration tested (15.62 µg/mL). In this case, the intracellular extracts were evaluated, but they did not exhibit relevant antioxidant effects.

#### 3.2.3. UV Protectant Effect

To assess the potential photoprotective effects of extracts on HaCat cells subjected to UVC-induced damage, their cytotoxicity was initially evaluated. HaCat cells treated with intracellular extracts at concentrations of 100 and 200 µg/mL exhibited moderate to strong cytotoxicity only for extracts obtained from *Bacillus* sp. 71, *Streptomyces* sp. 79, *Microbacterium* sp. 92, and *Micromonospora* sp. 93, reducing cell viability to below 70% (Figure 5A). Consequently, only the non-cytotoxic intracellular extracts were selected for photoprotective experiments (Figure 5B). Exposure of HaCat cells to UVC for 30 s in the absence of any protective compounds reduced cell viability to 55%. However, pre-treatment with 200 µM of Uvinul, a positive control, mitigated UVC-induced cell death, increasing cell viability to 85%. Based on the mentioned experimental setup, the extracts derived from *Pseudoalteromonas* sp. 39 exhibited the most significant UVC protective effect, enhancing cell viability to 76% at a concentration of 200 µg/mL, a level comparable to that of the positive control (Figure 5B). The same extract at 100 µg/mL also improved cell viability to 65%. In contrast, all other extracts demonstrated no substantial improvement in viability relative to UVC-treated cells without extract supplementation. In this case, the extracellular extracts were evaluated but they did not exhibit relevant photoprotective activity.

### 3.3. Genome-Based Identification, Genome Annotation, and Biosynthetic Potential Analysis of Selected Strains

Based on the above-mentioned results, *Pseudoalteromonas* sp. 39, *Streptomyces* sp. 79, *Microbacterium* sp. 92, and *Micromonospora* sp. 93 emerged as the best candidates to be considered for future coral restoration purposes and biotechnological applications. For this reason, these strains were selected for whole-genome sequencing and annotation. Summarized results of the general genomic features are shown in the table below (Table 2).

Analyses carried out through the GTDB-tk tool showed that three of the four selected strains were classifiable within the class Actinomycetia. The closest placements were found with *Streptomyces parvus* (ANI value 98.7%) for genome 79 and *Micromonospora arenicola* (ANI value 98.21%) for genome 93. The *Microbacterium* sp. 92 (ANI value < 95%) genome could not be assigned at species level, indicating its genomic and taxonomic novelty. Finally, strain 39 was phylogenetically close to *Pseudoalteromonas piscicida_B* (ANI value 98.7%). The phylogenomic analysis is displayed in Figure 6.

The four genomes were annotated using RAST, PROKKA, and DRAM to assess the genomic and metabolic potential in their beneficial role as symbionts able to support coral response to environmental stressors. AntiSMASH was used to validate the presence of relevant BGCs in the four genomes, potentially involved in the production of secondary metabolites. We targeted specific genes involved in oxidative and nitrogen stress response (ROS and RNS), and the strain’s ability to produce pigments and antimicrobial compounds [24,33,67]. The analysis summarized in Table 3 and Appendix A (and further complemented by Appendix A) reveals significant features linked to stress response and secondary metabolite biosynthesis in accordance with the detected bioactivities.

*P. piscicida* 39 revealed high potential for antioxidant activity and a UV protection effect. Its genome annotation revealed the presence of 92 genes involved in stress response, from which 38 are related to oxidative stress—in particular, genes that neutralize or reduce ROS (glutathione, superoxide dismutase, peroxidases, catalase). Regarding RNS, this strain shows genes necessary to cope with nitrosative stress (*nnrS* and *norR*). Moreover, *P. piscicida* 39 has lipoate regulatory protein YbeD and arylpolyene compounds, both of which are involved in antioxidant activity and hold more than eight NRPS (non-ribosomal peptide synthetase) clusters putatively encoding for novel bioactive peptides. Concerning *S. parvus* 79, a total of 12 genes corresponding to ROS and RNS response and the presence of pigments similar to those of strain 39 were detected. Furthermore, it showed terpene genes related to the production of carotenoids and numerous BGCs encoding for potentially novel antimicrobial compounds. On the other side, *Microbacterium* sp. 92 showed a low number of genes involved in the oxidative stress response. According to the antiSMASH analysis, it also encoded for terpene carotenoids and a limited number of antibiotic molecules. Finally, *M. arenicola* 93 held 41 genes linked to stress response, but only a small number were linked to oxidative stress. On the contrary, a diverse array of BGCs associated with secondary metabolite biosynthesis was observed by antiSMASH analysis. This strain was in fact characterized by the presence of nine T1PKS clusters, five NRPS clusters, and three terpene BGCs, suggesting its significant potential for polyketide and peptide-based antimicrobial production. Additionally, three lantipeptide class I, two lantipeptide class II, and siderophore-associated BGCs clusters were detected, along with a thiopeptide, a LAP cluster, and a T3PKS cluster. These results highlight and confirm its capacity for bioactive compound production, including potential novel antimicrobials. Furthermore, analyses carried out on the MEGARes databases highlighted that all genomes possessed genes for resistance to different antimicrobial compounds, including macrolides, aminoglycosides, and fluoroquinolones (Appendix A).

Inferences on the metabolic potential of each strain showed that, aside from the citrate cycle, the four strains could be potentially able to also utilize the pentose phosphate cycle and the glyoxylate cycle; some enzymes related to reductive metabolic pathways (e.g., Calvin cycle, Arnon–Buchanan cycle) were also found within their genomes, suggesting the potential for the uptake and utilization of CO_2_ within their metabolism. Few CAZymes could be reliably identified among the four strains, especially those targeting mixed-linkage glucans and amorphous cellulose (Appendix A). The presence of genes involved in metabolic pathways such as the Calvin–Benson–Bassham cycle and reverse TCA cycles, which would allow the organisms to utilize inorganic carbon, might represent an interesting and important adaptation to a host-associated lifestyle, exploiting the host’s respiratory processes to generate energy and store inorganic carbon as sugars for energy generation.

## 4. Discussion

This study explored the microbiomes, both at the molecular and the functional level, of two of the most abundant species of corals in Maldives, *Porites lobata* and *Acropora gemmifera*. The aims were (i) to highlight specific associations between microbiome composition and four coral eco-physiological conditions (i.e., small and large colonies, shallow and deep reef), (ii) to isolate cultivable microbial fraction and characterize the bioactivity, and (iii) to highlight the genome and metabolic content of selected strains for their potential environmental and human health applications.

To the best of our knowledge, no data are present in the literature on the microbiome characterization of these two coral species in Maldivian reefs. Thus, this work represents the first study describing associated microbial communities of *P. lobata* and *A. gemmifera* in that area combined with a preliminary bacterial isolation and characterization. *A. gemmifera* showed high consistency and similarity of microbial composition among the four conditions. In this coral species, the microbiome is predominated by the *Endozoicomonadaceae* family, both in shallow and in deep water. Different results were obtained from comparative analysis of *P. lobata* samples. In this case, microbial compositions changed along colony size and reef depth. This coral species seems to shape the symbiotic relationship with heterotrophic microorganisms during their growth. *Endozoicomonadaceae* was found in previous studies to be a common bacterial family associated with corals. Recent studies recorded this family in *Acropora* species collected in 99 reefs across the Pacific Ocean [68]. Colonies of *P. lobata* collected from six locations in Fouha Bay, Guam, also showed a predominance of *Endozoicomonadaceae* [69]. Within the *Endozoicomonadaceae* family, the genus *Endozoicomonas* demonstrated high resistance to the climate change-related stressors on *A. muricata* colonies (Taiwan) [70]. Also, the high percentage of Rhizobiaceae in *Porites lobata* microbiome has been reported in previous studies, as corals sampled in Caribbean reefs [71]. Among the most abundant families found in the surrounding environments, no similarity in microbial community composition was found between corals, water, and sediments. These findings are in accordance with previous investigations [72,73], highlighting how the main factors that shape microbial composition are coral species and growth stages, without a significant contribution from environmental matrices (i.e., seawater and sediments).

To elucidate the nature of the complex interactions occurring between corals and associated bacteria, the isolation of microorganisms in pure cultures is crucial. In our work, ten different bacterial genera were identified and screened for several bioactivities relevant for future coral restoration approaches and biotechnological purposes. Although the selected strains do not belong to the most abundant families observed in coral microbiomes, they demonstrated strong bioactivity and thus high relevance in coral ecological processes. Among them, the most active were identified as *Pseudoalteromonas piscicida* 39, *Streptomyces parvus* 79, *Microbacterium* sp. 92, and *Micromonospora arenicola* 93. The whole genome of these selected strains was sequenced, allowing for the identification, by phylogenomic analysis, of *Microbacterium* sp. 92 as novel species. The genus *Pseudoalteromonas* includes 48 described species (https://lpsn.dsmz.de/genus/pseudoalteromonas, accessed on 23 January 2025) and is known for its diverse ecological roles, including both pathogenicity and beneficial activity. For example, some species, like *Pseudoalteromonas piratica*, are implicated in coral diseases such as the white syndrome [74,75], whereas others produce bioactive compounds with antimicrobial, antioxidant, and antifouling properties [75,76]. In the work by Shnit-Orland et al., five out of six *Pseudoalteromonas* spp. isolated from different coral species showed antimicrobial activity against Gram-positive strains, especially *Bacillus cereus* and *Staphylococcus aureus* [74]. More recently, Ushijima et al. developed a potential probiotic treatment for infectious coral diseases in aquarium studies using *Pseudoalteromonas* sp. strain McH1-7. This bacterium produces at least two broad-spectrum antibacterials, korormicin and tetrabromopyrrole, along with other molecules (pseudoalterins) that are potentially active against many coral pathogens [77]. *P. shioyasakiensis* isolated from *Mussismilia braziliensis* detoxified radicals by producing ROS-reactive pigments, working as an antioxidant [78]. Similarly, three different *Pseudoalteromonas* symbionts of soft coral showed an antioxidant effect by producing carotenoid pigments [79]. Pigmented *Pseudoalteromonas* strains are particularly prolific in producing secondary metabolites, with their pigmentation often linked to antifouling and antibacterial activities [74]. The closest relative of *P. piscicida* 39 was isolated from hypersaline seawater in Mexico and is reported to produce bioactive compounds against the multidrug-resistant *Vibrio parahaemolyticus*, which causes high mortality in shrimps. In our study, the isolate 39 exhibited moderate antibacterial activity against *V. anguillarum*, *P. damselae*, and *S. aureus*. In addition, it demonstrated antioxidant activity and an orange pigmentation observed after approximately 24 h of incubation at 28 °C. In accordance with these results, the genome annotation of strain 39 showed numerous genes potentially involved in stress response, especially towards ROS and RNS and pigment production. AntiSMASH analysis of this strain revealed a diverse biosynthetic repertoire, including phosphonate, RiPP-like clusters, hybrid NRPS-like, arylpolyene, and T3PKS–NRPS hybrid BGCs. The limited similarity of these BGCs to known clusters suggests a strong potential for the discovery of novel bioactive metabolites. The other selected strains included in this study, *S. parvus* 79, *M. arenicola* 93, and the novel identified *Microbacterium* sp. 92, belong to Actinomycetota and are among the most prolific in terms of secondary metabolite production. For example, most of the drugs currently on the market derive from or are inspired by *Streptomyces*, so it is not surprising that the current isolate showed strong antimicrobial activity against most of the targeted strains [20,80,81]. A wide range of new molecules endowed with antimicrobial activity have already been discovered from coral-associated *Streptomyces*, confirming the enormous potential of these filamentous bacteria in the biotech industry [82]. In coral ecosystems, these bacteria contribute significantly to the health and resilience of their hosts. They are involved in processes such as nutrient cycling, protection against pathogens, and chemical signaling [83]. Specifically, *Streptomyces* strains produce bioactive compounds that act as antimicrobials, deterring opportunistic infections and maintaining the stability of the coral microbiome. Their production of antioxidants and UV-protectant molecules further underscores their importance in mitigating environmental stressors, such as oxidative damage and solar radiation [84]. The unique adaptations of these bacteria to the coral environment suggest that they may harbor unexplored pathways for the biosynthesis of natural products with unique structures and functions [85]. The novel *Microbacterium* sp. 92 exhibited antimicrobial activity towards 7 out of the 10 pathogens tested and was demonstrated to possess cryptic BGCs showing low similarity with known compounds. Currently, there are 156 species included in this genus (https://lpsn.dsmz.de/genus/microbacterium, accessed on 23 January 2025), and approximately 20% isolated from marine habitats [86]. These bacteria are integral members of the coral microbiome. Recent studies indicate that *Microbacterium* strains produce a range of bioactive compounds, including enzymes, antimicrobial agents, and antioxidants, which aid in mitigating environmental stressors and maintaining coral homeostasis [87].

Lastly, *M. arenicola* 93 was initially identified as *Salinispora arenicola*, changed after the update of bacterial nomenclature and taxonomy [88]. The genus Micromonospora is recognized as one of the most prolific producers of bioactive secondary metabolites. A review by Suqi Yan et al. (2022) reported 585 secondary metabolites associated with *Micromonospora*, categorizing them into eight major groups: aminoglycosides, macrolides, macrolactams, peptides, quinones, oligosaccharides, alkaloids, and miscellaneous compounds [89]. These bacteria thrive in diverse environments, including both terrestrial and marine ecosystems, and in association with corals. Within the family of Micromonosporaceae, with only nine species described (https://lpsn.dsmz.de/genus/salinispora, accessed on 23 January 2025) and despite being relatively understudied in coral ecosystems, the strictly marine genus *Salinispora* is widely recognized for its prolific production of secondary metabolites, exhibiting potent antimicrobial, anticancer, and anti-inflammatory properties, with potential applications in pharmaceuticals. Notable examples include salinosporamide A, a potent proteasome inhibitor currently in clinical trials for cancer therapy. The unique chemical structures of *Salinispora*-derived compounds make them attractive candidates for drug development and other industrial applications [90]. Within the coral environment, *Salinispora* strains could contribute to the protection of their host by producing natural products that inhibit the growth of pathogenic microorganisms (along with further unknown functions), thereby enhancing coral health and resilience [91]. Another interesting application for *Salinispora* was showed by the study of Becerril-Espinosa et al. (2022) [92]. The authors highlighted the potential of *S. arenicola* as a biostimulant for agriculture, promoting tomato plant growth and stress tolerance under saline and non-saline conditions. Notably, this obligated marine bacterium was able to establish an endophytic symbiosis, which induced an enhanced photosynthetic performance, regulated ROS formation, and induced the expression of salt-stress response genes such as SlHKT1;2. These findings highlight its potential as a sustainable tool to enhance crop resilience and productivity, while also providing insights into its possible role in supporting coral health status. The ability of *Salinispora arenicola* to thrive in nutrient-limited marine environments underscores its metabolic versatility and importance in sustaining coral ecosystems [93].

Although the genome analysis predicted the presence of many BGCs, their expression should be experimentally validated through metabolomics or heterologous expression. However, these findings support the fact that coral-associated microorganisms possess a remarkable capacity for secondary metabolite biosynthesis, with novel bioactive compounds being identified annually. Only between 2018 and 2022, over 385 novel compounds were identified from CAMs, both bacteria and fungi [94]. These microbes likely evolved the ability to produce diverse bioactive compounds to help corals combat predation, overgrowth, and fouling, among others. While the ecological roles of many of these compounds remain unclear, their pharmaceutical and industrial potential is significant. Coral-derived microbial metabolites exhibit diverse bioactivities, including antitumoral, antibacterial, antifungal, antifouling, anti-inflammatory, and antidiabetic properties, making them valuable for blue biotechnology and pharmaceuticals. This highlights the vast chemical diversity and economic potential of coral microbiomes. Our study represents a preliminary exploration of Maldivian CAMs in line with the concepts depicted in the roadmap for R&D on natural adaptation and assisted evolution of corals to climate change [95]. Following this direction, the identification of novel microbial symbionts and the induction of cryptic BGCs could indeed lead to the discovery of novel bioactive compounds [96] for both coral restoration and biotechnological applications.

## 5. Conclusions

This study showed the taxonomic, genomic, and functional diversity of coral-associated microorganisms and their critical roles in coral health and resilience. In the context of assisted evolution, leveraging beneficial CAMs for coral restoration involves enhancing these bacterial functions through targeted microbiome manipulation. By promoting microbial communities that confer stress tolerance, it is possible to improve coral’s overall resilience to climate change. Genetic selection and microbial inoculation may offer promising strategies to accelerate coral adaptation, ensuring its survival and continued ecological function in increasingly hostile ocean environments. The identification of relevant strains such as *S. parvus* 79, *M. arenicola* 93, and *P. piscicida* 39, as well as the novel species *Microbacterium* sp. 92, alongside the characterization of their biotechnological potential, provides a strong foundation to continue working on these strains to uncover their full potential. By addressing oxidative stress, nutrient imbalances, and pathogen defense, these bacteria represent promising tools for microbiome-based coral restoration strategies. Further exploration of coral microbiomes across diverse environmental conditions and experimental validation of these bacterial functions will enhance our understanding of host–microbe interactions under climate change scenarios.

Despite their ecological and biotechnological significance, our understanding of CAMs in coral ecosystems remains limited, and much remains to be uncovered about their role. Future research should aim to isolate novel strains, explore their functional and chemical diversity, and evaluate their contributions to both coral health and biotechnological innovation.

## Figures and Tables

**Figure 1 biology-14-00401-f001:**
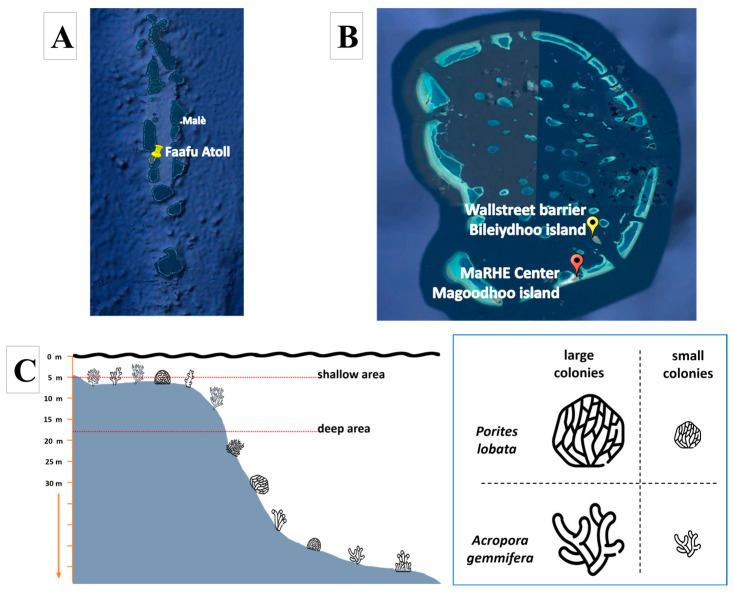
The Republic of Maldives with Faafu Atoll (**A**); Faafu Atoll, with MaRHE Center on Magoodhoo Island (red marker) and the sampling site named Wallstreet, in front of Bileiydhoo Island (yellow marker) (**B**). Sampling design (**C**). Two sampling areas were selected, a shallow area (5 m) and a deep area (17 m). In these areas, small (≤10 cm) and large colonies (≥30 cm) of *Porites lobata* and *Acropora gemmifera* were sampled in biological triplicate, for a total of 24 colonies.

**Figure 2 biology-14-00401-f002:**
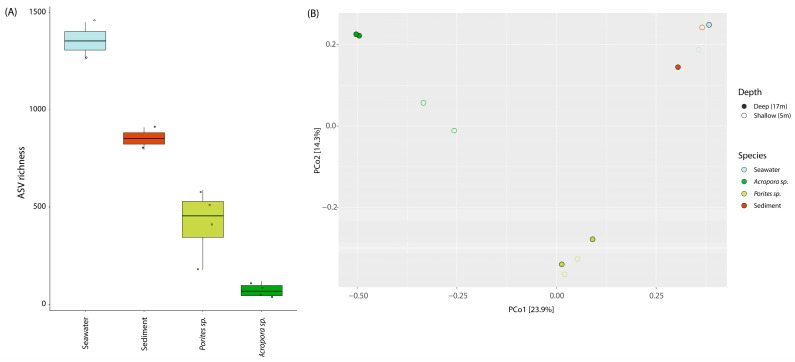
Boxplots showing the number of ASVs identified per sample type (**A**). PCoA plot showing the differences between samples (**B**).

**Figure 3 biology-14-00401-f003:**
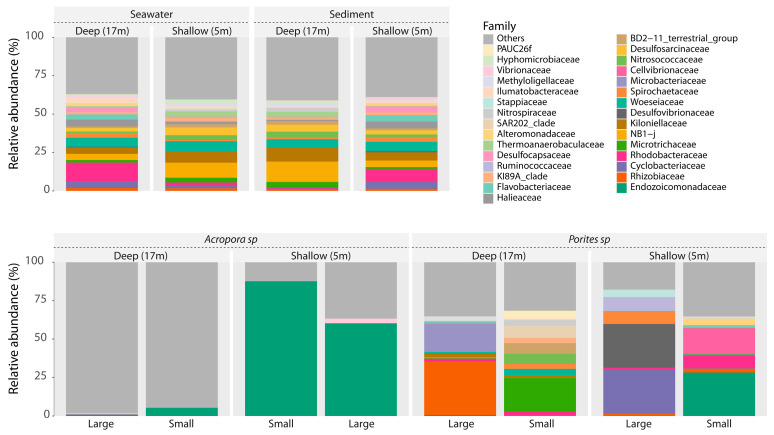
Bar plots showing the composition of prokaryotic assemblages associated with each sample.

**Figure 4 biology-14-00401-f004:**
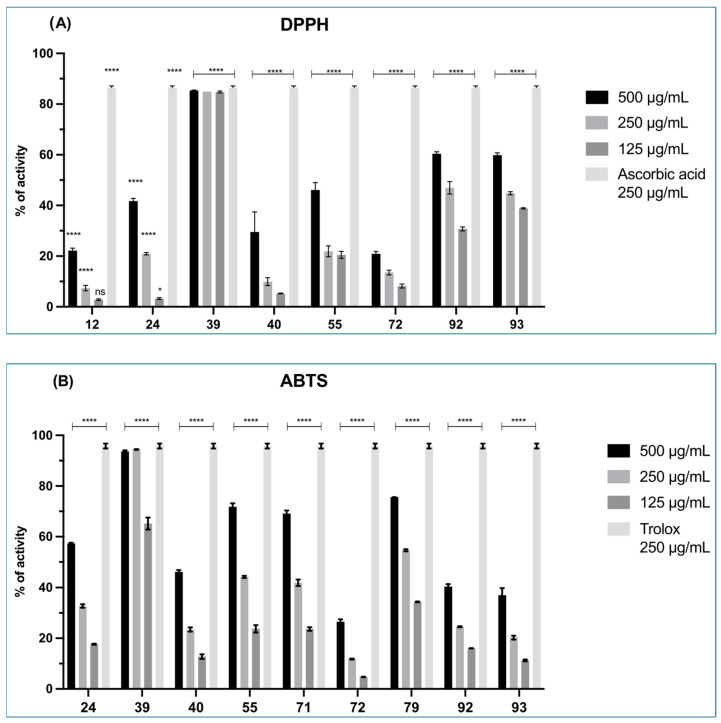
Mean % activity of DPPH (**A**) and ABTS (**B**) assays. Bars represent the mean ± SD (n = 3) from triplicate measurements for total extracellular extracts of the 10 selected strains. In the DPPH assay, ascorbic acid was used as the positive control (red dotted line). In ABTS assay, Trolox was used as the positive control (red dotted line). ANOVA ordinary 2-way Dunnett’s multiple comparisons: 0.1234 (ns), 0.0332 (*), <0.0001 (****).

**Figure 5 biology-14-00401-f005:**
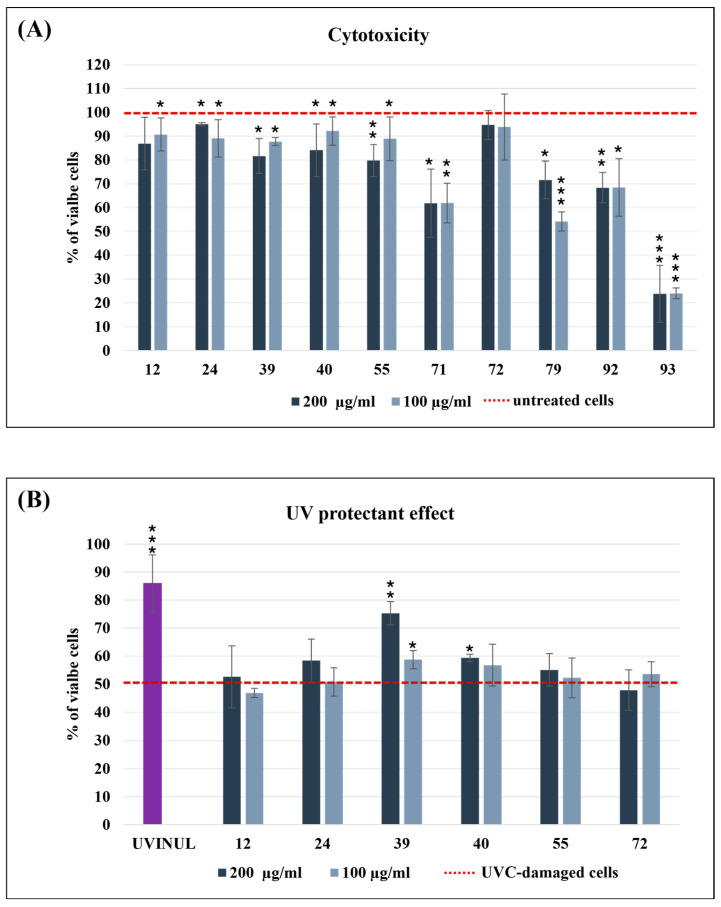
Cytotoxicity (**A**), cell viability assay on HaCat cells. Bar graphs represent the percentage of viable cells after 24 h of treatment with intracellular extracts; control (untreated cells) represents 100% viability (red dotted line). UVC protectant effect (**B**), assay of intracellular extracts on HaCat viability cells injured with UVC. The negative control (cells not exposed to UVC) represents 100% of viable cells, whereas the positive control (cells only exposed to UVC) is reported as the red dotted bar. Bar graphs represent the percentage of viable cells after UV injury and 24 h of treatment with extracts. Uvinul was used at 200 µM as the positive control. Assays were performed in biological triplicate, and the graphs represent means ± standard deviations. The *p*-values are reported with asterisks: 0.05 (*), 0.001 (**), 0.0001 (***).

**Figure 6 biology-14-00401-f006:**
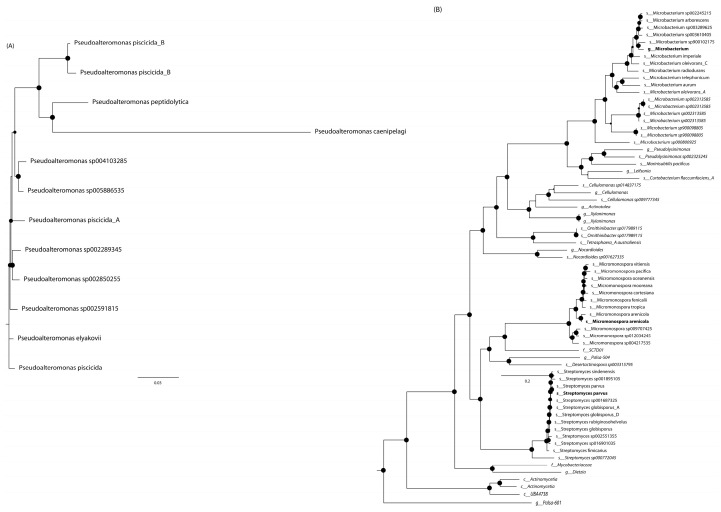
Phylogenetic trees of coral-associated bacterial strains. For each strain, the most appropriate HMM set was chosen according to its taxonomic affiliation: (**A**) Gammaproteobacteria for genome 39 and (**B**) Actinobacteria for the others. The newly selected strains are shown in bold text, while genomes collected from environmental databases are shown in italics. The circle size at each branching refers to the branching support.

**Table 1 biology-14-00401-t001:** Antimicrobial activity reporting the MIC (µg/mL) of the most active extracts towards the pathogen panel. Values of up to 500 µg/mL were reported; values ≤ 125 µg/mL were considered promising inhibition activities.

Pathogen Strains	Active Strains	MIC (µg/mL)
*E. coli* ATCC 10536	79	500
*L. monocytogenes* MB 677	79	1
	92	1
	93	0.5
*P. damselae* subps. *piscicida* ATCC 51736	39	30
	79	0.0098
	92	125
	93	125
*S. aureus* methicillin resistant	39	500
	79	1
	92	125
	93	31
*S. aureus*—quinolone resistant	79	500
	92	2
	93	1
*S. aureus*—macrolide resistant	79	0.5
	92	2
	93	20
*S. aureus* 6538p	39	125
	79	0.3
	92	0.313
	93	1.25
*V. anguillarium* ATCC 19264	39	0.031
	92	20
	93	60
*C. albicans* ATCC 76485	79	0.25
*S. aureus*—vancomycin resistant	-	-
*P. aeruginosa* PAO1	-	-

**Table 2 biology-14-00401-t002:** General features of the whole genome of the 4 selected strains.

	*Pseudoalteromonas piscicida* 39	*Streptomyces* *parvus* 79	*Microbacterium* sp. 92	*Micromonospora* *arenicola* 93
Completeness (%)	100	99.97	99.96	100
Redundancy (%)	0.1	0.1	0,13	1.16
Coding density	0.88	0,89	0.92	0.87
Genome size (bp)	5,096,534	8,193,082	3,204,941	5,583,533
GC content (%)	0.43	0.72	0.7	0.7
Total coding sequences	4400	7126	3032	4983
Total contigs	45	102	7	55
tRNAs	64	69	46	54

**Table 3 biology-14-00401-t003:** Summary table of RAST results on genes involved in the response to oxidative and nitrogen stress present in strains 39, 79, 92, and 93. For each strain, “**1**” indicates the presence of the gene and “**0**” its absence. SOD refers to one the following: Mn, Fe, Cu-Zn, nickel-dependent, Mn/Fe, Fe-Zn. “Nitrosative stress” refers to one of the following: NnrS, NorR, flavorubredoxin, NsrR.

	*Pseudoalteromonas piscicida* 39	*Streptomyces* *parvus* 79	*Microbacterium* sp. 92	*Micromonospora arenicola* 93
Glutathione biosynthesis	1	1	1	1
β-carotene	0	1	0	1
Mannitol synthesis	0	0	1	1
DMS	0	0	0	0
DMSP	0	0	0	0
Peroxinitrite reduction (ahpC)	1	0	0	0
Nitric oxide reductase	1	0	0	0
Lipoic acid metabolism	1	1	1	0
Catalase-peroxidase (KatG)	1	1	1	1
Catalase (KatE)	1	1	1	1
SOD	1	1	1	1
Nitrosative stress	1	1	0	0

## Data Availability

All data obtained in this study were reported in the main text or in Appendix A; if readers needs more information and details can contact corresponding authors.

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
