# Peer review of "Novel Insights and Genomic Characterization of Coral-Associated Microorganisms from Maldives Displaying Antimicrobial, Antioxidant, and UV-Protectant Activities"

_biology, 2025, doi:10.3390/biology14040401_

Round 1
Reviewer 1 Report
Comments and Suggestions for Authors
The paper "New Insights and Genomic Characterization of Maldivian Coral-Associated Microorganisms with Antimicrobial, Antioxidant, and UV-Protective Activities" provides valuable information on the coral-associated bacterial microbiota and its role in reef health and resilience. Furthermore, the identification and characterization of bacterial strains with antimicrobial, antioxidant, and UV-protecting properties highlights their potential for both coral restoration in threatened ecosystems and the development of bioactive compounds with potential biotechnological applications.
However, I believe some aspects of the study could be improved.
Introduction:
The citations by Rohwer et al., 2002; Knowlton and Rohwer, 2003, should be included within the holobiont concept.
Include the relevance of combining the use of culture-dependent and metagenome techniques in the present study.
Adding the hypothesis of the study and what it aims to demonstrate.
Materials and methods:
Amount of sediment used for the study
Specify whether the DNA from the coral is a product of the mixture of tissue, mucus, and skeleton or if it is only from the tissue
Results:
I believe very few ASVs were obtained; please clarify in the materials and methods and results.
Discussion:
should address why the microbiota was different between water sediment and corals and delve into the genera that showed bioactivity and their implications for coral health.
Citation 65 is incorrect; please replace it with: Hernández-Zulueta, Joicye, Leopoldo Díaz-Pérez, Alex Echeverría-Vega, Gabriela Georgina Nava-Martínez, Miguel Ángel García-Salgado, and Fabián A. Rodríguez-Zaragoza. 2023. "An Update of Knowledge of the Bacterial Assemblages Associated with the Mexican Caribbean Corals Acropora palmata, Orbicella faveolata, and Porites porites" Diversity 15, no. 9: 964. https://doi.org/10.3390/d15090964
Author Response
REVIEWER 1
The paper "New Insights and Genomic Characterization of Maldivian Coral-Associated Microorganisms with Antimicrobial, Antioxidant, and UV-Protective Activities" provides valuable information on the coral-associated bacterial microbiota and its role in reef health and resilience. Furthermore, the identification and characterization of bacterial strains with antimicrobial, antioxidant, and UV-protecting properties highlights their potential for both coral restoration in threatened ecosystems and the development of bioactive compounds with potential biotechnological applications.
However, I believe some aspects of the study could be improved.
Introduction:
The citations by Rohwer et al., 2002; Knowlton and Rohwer, 2003, should be included within the holobiont concept.
We thank the reviewer for this suggestion; we added the new citations as references 15 and 16.
Include the relevance of combining the use of culture-dependent and metagenome techniques in the present study.
We included the following sentence clarifying the importance of using this double approach (line 139-141): The combination of both culture-dependent and metabarcoding approaches could provide more informative data increasing the current knowledge about CAMs and their importance in coral conservation.
Adding the hypothesis of the study and what it aims to demonstrate.
We thank the reviewer for this suggestion; we added hypothesis and aims in the final part of the introduction section:
- Lines 142-145: This research started from the scientific hypothesis that the bio-chemical complexity of the symbiotic relationship between corals and microorganisms can be a rich source of bacteria already chemically adapted to face with climate change related stressors. Thus, this study aimed to use a holistic approach to investigate…
- Lines 161-163: Thus, the final aim of this work is the identification of promising bacteria with associated information on bioactivity and genome potential that could lay the basis for…
Materials and methods:
Amount of sediment used for the study
Specify whether the DNA from the coral is a product of the mixture of tissue, mucus, and skeleton or if it is only from the tissue
We added some information in the section 2.2 as follow: Sediments and corals (300 mg) were extracted using DNeasy PowerSoil Pro Kit (Qiagen, cat. n. 47014). Corals (composing by a mixture of tissue, mucus and skeleton) were fragmented using sterile pestle and mortar prior to DNA extraction.
Results:
I believe very few ASVs were obtained; please clarify in the materials and methods and results.
We thank the reviewer for this comment; we improved description of methods and results. A total of 4635 ASVs were retained after rarefaction of the feature table; the ranges of richness values obtained are in the range for other coral-related bacterial microbiomes, such as in our previous study on Corallium rubrum from the Mediterranean Sea (e.g. Corinaldesi et al., 2022, https://doi.org/10.1016/j.scitotenv.2022.153701)."
Discussion:
should address why the microbiota was different between water sediment and corals and delve into the genera that showed bioactivity and their implications for coral health.
We thank the reviewer for this suggestion, and we modified section 4, adding discussion on differences among matrices (corals, water and sediments), lines 655-660: Among the most abundant families found in the surrounding environments, no similarity in microbial community composition was found between corals, water and sediments. These findings are in accordance with other investigations, highlighting how microbial composition is highly shaped among coral species and growth stages and without a significant contribution from environmental matrices (i.e., seawater and sediments).
We did not find a correlation between the most abundant families of coral microbiomes and bioactive strains. Thus, we reported this information in a sentence at lines 664-667: Although the selected strains do not belong to the most abundant families observed in coral microbiomes, they demonstrated strong bioactivity and thus high relevance in coral ecological processes.
Citation 65 is incorrect; please replace it with: Hernández-Zulueta, Joicye, Leopoldo Díaz-Pérez, Alex Echeverría-Vega, Gabriela Georgina Nava-Martínez, Miguel Ángel García-Salgado, and Fabián A. Rodríguez-Zaragoza. 2023. "An Update of Knowledge of the Bacterial Assemblages Associated with the Mexican Caribbean Corals Acropora palmata, Orbicella faveolata, and Porites porites" Diversity 15, no. 9: 964. https://doi.org/10.3390/d15090964
We modified this reference
Reviewer 2 Report
Comments and Suggestions for Authors
The ms presented explores the CAM's biotechnological potential. This research is highly scientifically sound for the scientific community and to determine coral resistance and for future coral restoration initiatives. The manuscript is well written and i only have few minor suggestions.
Introduccion section. The authors describe that the results are also applicable for further human biotechnological applications, however, this is unclear in the discussion section. Therefore, it is suggested that either the general scope be modified or a clear discussion be included on the potential for use in human applications.
Sampling site. It would be relevant to include a brief description of the environmental characteristics of the site, as well as the time of collection. Also, the general coral structure in the area, which criteria was used to divide small (-10 cm) from large colonies?
Materias and methods section
Coral collection. Please include details on the reagents and materials used (e.g. RNA later-invitrogen?, cellulose filters .. watmann?), for further protocol replicability.
Section. 2.6 Include the levels of UVC used to induce damage.
Bacterial cultivation. Which criteria (morphological and phylogenetic) were used to select the bacterial strains?
Antimicrobial assay. Please include the permit for handling pathogen strains in the proper section and details of how they were obtained/isolated.
Results
Line 370. Which criteria (any range for small vs large?? etc) was used to identify the characterization of the colonies?
General comment for the figures. Increase/change the font size as in the current version the legends, axes or references cannot be read
Figure 7. Is suggested to include the figure in supplemental material.
Author Response
REVIEWER 2
The ms presented explores the CAM's biotechnological potential. This research is highly scientifically sound for the scientific community and to determine coral resistance and for future coral restoration initiatives. The manuscript is well written and i only have few minor suggestions.
Introduction section. The authors describe that the results are also applicable for further human biotechnological applications, however, this is unclear in the discussion section. Therefore, it is suggested that either the general scope be modified or a clear discussion be included on the potential for use in human applications.
We thank the reviewer for this comment. We integrated into the discussion section the relevance of our study in the context of blue biotechnology and pharmaceuticals, which is already a hot topic in coral-microbes bioprospecting.
Sampling site. It would be relevant to include a brief description of the environmental characteristics of the site, as well as the time of collection.
We added a short description of the sampling site in the 2.1 section, lines 176-185: Wallstreet is characterized by high coral coverage and diversity interspersed with small sand pockets and distinguished by a broad, coral-rich flat that extends 20 meters wide. A steep wall descends to depths around 30 meters, gradually sloping to 35 m before transitioning into the sandy seabed. This vertical descent features various crevices, caves, and overhangs that provide essential shelter for a diverse array of marine life. The site is typically influenced by medium to slow currents, fostering optimal conditions for the proliferation of diverse coral and fish communities.
Coral colonies were identified and simultaneously sampled at approximately 10 am to ensure that all replicates were subjected to the same environmental conditions (e.g. light, temperature, current), following the scheme illustrated in Figure 1C.
Also, the general coral structure in the area, which criteria was used to divide small (-10 cm) from large colonies?
The rationale for selecting both small and large colonies lies in the aim to compare the microbiomes of corals at different phases of their life cycle. As outlined in the manuscript (Page 4, lines 145-147), “this study aimed to use a holistic approach to investigate: i) how and whether different depths and growth stages drive the structure of coral microbiomes…”). Consequently, we sought to clearly differentiate between larger, adult colonies and smaller, juvenile ones.
To achieve this distinction, we selected colonies with a minimum size difference of 20 cm in diameter for each species (juvenile/small: < 10 cm; adult/large: > 30 cm), ensuring that the colonies compared were indeed at different intraspecific growth stages. While we recognize that growth rates and the minimum size for reproduction can vary across coral species, it is generally accepted that corals less than 10 cm in diameter are considered young in relation to the longevity of most coral species, as suggested by Birkeland (book title: Life and Death of Coral Reefs;1997), and that colonies smaller than 5 cm in diameter are typically classified as "recruits". Moreover, several studies have cited 10 cm in diameter as a threshold size to differentiate juvenile colonies from adults (see, for example, Cardini et al., 2012, https://doi.org/10.1017/S0025315411001561; Montalbetti et al., 2019, DOI: 10.1007/s10750-018-3786-6; Montalbetti et al., 2022, https://doi.org/10.1007/s00338-022-02230-1; Norström et al., 2007, https://doi.org/10.1007/s00227-006-0458-2).
We better specified in the text this concept, see section 2.1, lines 185-189: In detail, two sampling depths were identified at 5 m (shallow) and 17 m (deep). In both depths, small (< 10 cm in diameter, juvenile colonies) and large colonies (> 30 cm in diameter, adult colonies) of Porites lobata and Acropora gemmifera were sampled…
Materias and methods section
Coral collection. Please include details on the reagents and materials used (e.g. RNA later-invitrogen?, cellulose filters .. watmann?), for further protocol replicability.
We thank the reviewers for this comment. We improved the method description with these details.
Section. 2.6 Include the levels of UVC used to induce damage.
We added more technical information on the UV lamp used.
Bacterial cultivation. Which criteria (morphological and phylogenetic) were used to select the bacterial strains?
We thank the reviewer for this observation. We isolated the strains according to their different morphology, but we selected and cultivated ten of them based on their different taxonomy. We clarified it in the section 2.5.
Antimicrobial assay. Please include the permit for handling pathogen strains in the proper section and details of how they were obtained/isolated.
We thank the reviewer for this comment; we added in the acknowledgment section colleagues and department that kindly provided these strains. Our institutions have facilities and documentation for handling and performing antimicrobial assay using strains reported in this manuscript.
Results
Line 370. Which criteria (any range for small vs large?? etc) was used to identify the characterization of the colonies?
See the third answer we provided, since it contains information for this comment.
General comment for the figures. Increase/change the font size as in the current version the legends, axes or references cannot be read
We modified all figures to improve readability.
Figure 7. Is suggested to include the figure in supplemental material.
We agreed with this recommendation; thus, we moved figure 7 from main text to supplemental materials.
Reviewer 3 Report
Comments and Suggestions for Authors
The manuscript "Novel Insights and Genomic Characterisation of Coral-Associated Microorganisms from Maldives displaying Antimicrobial, Antioxidant, and UV-Protectant Activities" by Esposito et al. presents a study on the microbiome of two coral species from the Maldives, Porites lobata and Acropora gemmifera, focusing on the bioactivity of isolated coral-associated bacteria (CAMs). The authors characterize microbial communities using metabarcoding and cultivation techniques and evaluate their antimicrobial, antioxidant, and UV-protectant properties. Furthermore, the genomes of four promising strains (Pseudoalteromonas piscicida 39, Streptomyces parvus 79, Microbacterium sp. 92, Micromonospora arenicola 93) were sequenced and analyzed for secondary metabolite biosynthetic gene clusters.
The study provides valuable insights into coral-microbe interactions and the potential of CAMs for coral restoration and biotechnological applications. The manuscript is well-written and presents an important topic in environmental microbiology and biotechnology. However, there are few aspects that require clarification and improvement, listed hereafter.
- The introduction provides a general overview of coral-associated microorganisms (CAMs) and their potential roles in coral health and resilience, but it lacks depth in contextualizing the study within the broader scope of coral microbiome research and biotechnology. While it briefly mentions the threats faced by coral reefs and the relevance of CAMs, it does not fully articulate the ecological or conservation significance of studying Porites lobata and Acropora gemmifera specifically. A clearer justification for selecting these species—such as their role as reef builders, their differential responses to environmental stress, or their potential as models for microbial symbiosis—would enhance the study’s framing. Additionally, the introduction should better integrate existing knowledge on microbial-based coral restoration strategies, such as the concept of Beneficial Microorganisms for Corals (BMCs) and previous probiotic interventions, to position the study’s objectives within ongoing efforts in the field. Furthermore, while the paper highlights the biotechnological potential of CAMs, it would benefit from a more explicit discussion of how microbial bioactivity screening aligns with existing drug discovery and environmental applications. Strengthening these aspects would provide a more compelling rationale for the study and clarify how its findings contribute to both fundamental and applied research in marine microbiology and conservation.
- The description of sequencing and bioinformatics analyses needs more clarity, particularly:
- The specific 16S rRNA primers used.
- The rarefaction depth of the ASV table.
- What was the quality threshold for sequence filtering?
- The authors incubated plates for "up to 3 months" for slow-growing strains. However, this is an unusually long time and may allow for contamination. Was there any quality control or repeated sub-culturing to confirm strain purity?
- For antioxidant and UV-screening assays, it would be helpful to clarify if the observed bioactivity was statistically significant compared to controls.
- The cytotoxicity assay on HaCaT cells lacks a dose-response curve. Were other mammalian cell lines tested for general toxicity?
- The genome annotation and biosynthetic gene cluster (BGC) prediction using antiSMASH are strong points of the study. However, the authors do not discuss potential limitations of genome mining approaches. Were any of the predicted BGCs validated experimentally, e.g., through metabolomics or heterologous expression?
The Microbacterium sp. 92 strain is suggested to be novel, but there is no mention of ANI (Average Nucleotide Identity) or dDDH (digital DNA-DNA hybridization) analysis to confirm its novelty. These should be reported if available.
-The study suggests that the selected bacterial strains could be used in coral restoration efforts. However:
How would these bacteria be introduced to corals in a real-world setting?
Do they persist in the coral microbiome long-term, or would repeated inoculation be required?
Have any field trials or mesocosm experiments been attempted?
Could introducing these strains have unintended ecological consequences, such as disrupting native microbiota?
Minor comments
-The abstract is informative but somewhat dense. Consider simplifying the last few sentences to improve readability.
Figures and tables:
- In Table 1 (antimicrobial activity), the authors should clarify the MIC (µg/mL) values with a footnote explaining the significance threshold for antimicrobial effectiveness.
- Supplementary file, last heatmap: legend is upside down
Author Response
REVIEWER 3
The manuscript "Novel Insights and Genomic Characterisation of Coral-Associated Microorganisms from Maldives displaying Antimicrobial, Antioxidant, and UV-Protectant Activities" by Esposito et al. presents a study on the microbiome of two coral species from the Maldives, Porites lobata and Acropora gemmifera, focusing on the bioactivity of isolated coral-associated bacteria (CAMs). The authors characterize microbial communities using metabarcoding and cultivation techniques and evaluate their antimicrobial, antioxidant, and UV-protectant properties. Furthermore, the genomes of four promising strains (Pseudoalteromonas piscicida 39, Streptomyces parvus 79, Microbacterium sp. 92, Micromonospora arenicola 93) were sequenced and analyzed for secondary metabolite biosynthetic gene clusters.
The study provides valuable insights into coral-microbe interactions and the potential of CAMs for coral restoration and biotechnological applications. The manuscript is well-written and presents an important topic in environmental microbiology and biotechnology. However, there are few aspects that require clarification and improvement, listed hereafter.
- The introduction provides a general overview of coral-associated microorganisms (CAMs) and their potential roles in coral health and resilience, but it lacks depth in contextualizing the study within the broader scope of coral microbiome research and biotechnology. While it briefly mentions the threats faced by coral reefs and the relevance of CAMs, it does not fully articulate the ecological or conservation significance of studying Porites lobata and Acropora gemmifera specifically. A clearer justification for selecting these species—such as their role as reef builders, their differential responses to environmental stress, or their potential as models for microbial symbiosis—would enhance the study’s framing.
The decision to focus on Porites lobata and Acropora gemmifera is primarily due to their high abundance both in the study area and throughout the Maldives, as extensively documented in various studies (Pichon & Benzoni, 2007, https://doi.org/10.11646/zootaxa.1441.1.2; Seveso et al., 2018, 10.1007/s00338-018-1703-0; Jimenez et al. 2021, https://doi.org/10.1016/j.seares.2012.04.011). These coral species are representative of the Maldivian reef ecosystem and play essential ecological roles from several perspectives. Moreover, as noted by the Reviewer, it is well established that these two species exhibit notable morphological differences—Porites lobata has a massive growth form, while Acropora gemmifera has a branched growth form. They also differ in terms of tissue and skeleton thickness, as well as at the molecular, physiological, and metabolic levels. These distinctions contribute to their varying susceptibility and tolerance to environmental stress, particularly the stressors associated with climate change that lead to coral bleaching. Generally, the genus Porites is considered more resistant to bleaching, although showing a slow recovery due to their slow growth rate, while Acropora is highly bleaching susceptible. However, Acropora has a higher growth rate and shows greater resilience and ability to recover following environmental stressors (see Loya et al., 2001, https://doi.org/10.1046/j.1461-0248.2001.00203.x; Guest et al., 2012, doi: 10.1371/journal.pone.0033353; Suggest and Smith, 2020, doi: 10.1111/gcb.14871; van Woesik et al 2022, doi: 10.1111/gcb.16192). This different behavior and response were also evident during past bleaching events in the Maldives (Ibrahim et al., 2017, Status of coral bleaching in the Maldives 2016; Cowburn et al., 2019, doi: 10.3354/meps13044; Pisapia et al., 2019, https://doi.org/10.1038/s41598-019-44809-9). Given these differences, it is also particularly interesting to investigate the potential role of the microbiome associated with these two coral genera in influencing their responses to environmental changes. We reported some of this information into the introduction section (see lines 151-158).
Additionally, the introduction should better integrate existing knowledge on microbial-based coral restoration strategies, such as the concept of Beneficial Microorganisms for Corals (BMCs) and previous probiotic interventions, to position the study’s objectives within ongoing efforts in the field.
We thank the reviewer for this comment. We integrated in the introduction section the main microbial-based therapies with the related references, which support the study conducted in our manuscript.
Furthermore, while the paper highlights the biotechnological potential of CAMs, it would benefit from a more explicit discussion of how microbial bioactivity screening aligns with existing drug discovery and environmental applications. Strengthening these aspects would provide a more compelling rationale for the study and clarify how its findings contribute to both fundamental and applied research in marine microbiology and conservation.
We thank the reviewer for this comment. We integrated into the discussion section the relevance of our study in the context of blue biotechnology and pharmaceuticals, which is already a hot topic in coral-microbes bioprospecting.
- The description of sequencing and bioinformatics analyses needs more clarity, particularly:
- The specific 16S rRNA primers used.
- The rarefaction depth of the ASV table.
- What was the quality threshold for sequence filtering?
We added all the information suggested by the reviewer.
- The authors incubated plates for "up to 3 months" for slow-growing strains. However, this is an unusually long time and may allow for contamination. Was there any quality control or repeated sub-culturing to confirm strain purity?
Yes, it is a long time, however especially during the isolation phase, some strains could require more time to adapt to the new environment and grow. Here, all the colonies were collected, re-streaked in fresh agar plate, and if pure, stored in glycerol stock at -80 °C and identified. We clarified this in the text, see section 2.4, line 257.
- For antioxidant and UV-screening assays, it would be helpful to clarify if the observed bioactivity was statistically significant compared to controls.
We performed statistical analysis for these three assays and we modified the figure 4 and 5 accordingly.
- The cytotoxicity assay on HaCaT cells lacks a dose-response curve. Were other mammalian cell lines tested for general toxicity?
We did not test total extracts on other human cell lines. Working with total extracts has the enormous advantage of performing quick and informative assays, since it excludes long and complex fractionation and purification steps. But results should be considered a preliminary insight into potential bioactivity. On the contrary, dose-response curve and mechanisms of action are experimental steps mostly suitable for enriched fractions and/or purified compounds, since bioactive molecule/number of cells ratio can be easily controlled and modulated.
- The genome annotation and biosynthetic gene cluster (BGC) prediction using antiSMASH are strong points of the study. However, the authors do not discuss potential limitations of genome mining approaches. Were any of the predicted BGCs validated experimentally, e.g., through metabolomics or heterologous expression?
We thank the reviewer for this comment. Very often BGCs can be cryptic or silent and their induction may be affected by different factors, especially the cultivation conditions (doi:10.3390/ijms22169055). In this work we adopted only one cultivation medium to obtain preliminary data. Currently, we are already applying the OSMAC (One Strain Many Compounds) approach coupled with untargeted metabolomics to produce novel bioactive molecules, that could be part of a next research paper. As suggested by the reviewer we inserted in the last paragraph of the Discussion a sentence highlighting the limitation in the BGCs analysis (line 754).
The Microbacterium sp. 92 strain is suggested to be novel, but there is no mention of ANI (Average Nucleotide Identity) or dDDH (digital DNA-DNA hybridization) analysis to confirm its novelty. These should be reported if available.
We evaluated both ANI and dDDH. We inserted the ANI values in section 3.3.
The study suggests that the selected bacterial strains could be used in coral restoration efforts. However:
How would these bacteria be introduced to corals in a real-world setting?
Do they persist in the coral microbiome long-term, or would repeated inoculation be required?
Have any field trials or mesocosm experiments been attempted?
Could introducing these strains have unintended ecological consequences, such as disrupting native microbiota?
These are interesting scientific questions that we are trying to answer and hopefully will be part of a next research articles. We are now testing how addition of single or mixed bioactive strains to coral fragments in small tanks in controlled conditions can stably modified the microbiome composition. Aim of this ongoing experimental step is the understanding of conservation of bioactivity from in vitro models (e.g., human cells) in vivo models (e.g., corals). Long term effects and field experiments will be downstream activities that need strong scientific data before to be set up. Thus, we believe that this paper represent a first group of data for a possible use of bioactive bacteria for field applications.
Minor comments
-The abstract is informative but somewhat dense. Consider simplifying the last few sentences to improve readability.
We thank the reviewer for this suggestion; we eliminated some superfluous information in the last sentences.
Figures and tables:
- In Table 1 (antimicrobial activity), the authors should clarify the MIC (µg/mL) values with a footnote explaining the significance threshold for antimicrobial effectiveness.
We added it to the legend Table.
- Supplementary file, last heatmap: legend is upside down
We modified the figure.